# Prevalence, patterns and predictors of metabolic abnormalities in Nigerian hypertensives with hypertriglyceridemic waist phenotype: A cross sectional study

Casmir E. Amadi[1]*, Amam C. Mbakwem[1], Dolapo C. Duro[2], Ifeoma C. Udenze[3], Clement M. Akinsola[4], Jayne N. Ajuluchukwu[1], David A. Wale Oke[1]

1 Department of Medicine, College of Medicine, University of Lagos, Lagos, Nigeria, 2 Leverage Consulting Limited Abuja, Abuja, Nigeria, 3 Department of Chemical Pathology, College of Medicine, University of Lagos, Lagos, Nigeria, 4 Department of Community Health, College of Medicine, University of Lagos, Lagos, Nigeria

* acetalx@yahoo.com

**Data Availability Statement:** Data are within the Supporting information files.

## Abstract

### Background

Simultaneous presence of elevated waist circumference and hypertriglyceridemia (HTGW) is a simple and low-cost measure of visceral obesity, and it is associated with a plethora of cardio-metabolic abnormalities that can increase the risk of cardiovascular diseases and incident Type 2 diabetes mellitus. We decided to study the prevalence, patterns, and predictors of metabolic abnormalities in Nigerian hypertensives with the HTGW phenotype.

### Methods

The medical records of 582 hypertensives with complete data of interest were retrieved and analyzed for the study. Their socio-demographic data, anthropometric data, and booking blood pressure values were retrieved. The results of their fasting plasma glucose, lipid profile, uric acid and serum creatinine were also retrieved for analysis.

### Results

The mean age of the study population was 56.2 ±13.6, with 53.1% being males. The prevalence of smoking and use of alcohol was 4.3% and 26.5% respectively. The prevalence of the HTGW phenotype was 23.4% and were predominantly males (61%). Subjects with the HTGW phenotype were more obese assessed by waist circumference (WC) and body mass index (BMI). Mean serum total cholesterol, triglyceride, very low-density lipoprotein, uric acid, and creatinine were significantly higher in the HTGW phenotype (p = 0.003; <0.001; <0.001; 0.002 and <0.001 respectively). The prevalence of newly diagnosed Type 2 diabetes was 28.7%. There was also a preponderance of cardio-metabolic abnormalities (obesity, dyslipidaemia, hyperuricemia) in the HTGW phenotype. In both males and females, the

**Funding:** The authors received no specific funding for this work.

**Competing interests:** The authors have declared that no competing interests exist.

HGTW phenotype was significantly associated with elevated Tc, TG, VLDL, hyperuricemia and atherogenic index of plasma.

## Conclusion

The HTGW phenotype is common amongst Nigerian hypertensives, and it is associated with metabolic abnormalities.

## Introduction

Cardiovascular disease (CVD) remains the number one cause of preventable and premature death globally, accounting for 17.9 million deaths in 2019 [1]. The Low-Middle Income Countries (LMICs) bear about 80% of this burden due to their enormous population, rapid economic development, lifestyle changes, and a high prevalence of co-occurring CVD risk factors [2, 3]. Obesity (diagnosed either as elevated Body Mass Index or elevated waist circumference) is a traditional risk factor for CVD. Body mass index (BMI) is commonly used to assess general obesity. However, waist circumference (WC), as a measure of abdominal obesity, has been proven to be a better predictor of CVD than BMI [4–6]. However, use of WC alone is limited by the fact that it is not able to discriminate between intra-abdominal/visceral adiposity and subcutaneous abdominal adiposity. Several studies have shown that visceral adiposity rather subcutaneous abdominal adiposity is plays a crucial role in glucose metabolism, blood pressure homeostasis, lipid metabolism and inflammation and has positive associations with metabolic abnormalities, including hypertension, type 2 diabetes, dyslipidemia, and hyperinsulinemia [7, 8]. Measurements of visceral adiposity require complex and expensive imaging techniques such as computed tomography (CT) or magnetic resonance imaging (MRI) and are thus not suitable and cost effective for the general population and for routine clinical evaluation of visceral adiposity. Lemieux et al. [9] demonstrated that the hypertriglyceridemic waist (HTGW) phenotype (defined as co-occurring hypertriglyceridemia and elevated WC) was a simple and inexpensive tool to identify cardiometabolic abnormalities (especially the metabolic triad of hyperinsulinemia, elevated Apolipo B and small dense LDL-c) in individuals with visceral obesity in routine clinical encounters with patients [10, 11]. HTGW is also associated with increased risk of angiography confirmed coronary artery disease and sub-clinical atherosclerosis [12, 13].

Hypertensives have been reported to be at increased of risk having HTGW phenotype [14–16]. In these hypertensives this phenotype is associated with a plethora of metabolic abnormalities, which increases the risk of incident CVD [17, 18]. In Nigeria the prevalence of hypertension is about 30.8% [19]. Often hypertension aggregates other CVD risk factors especially visceral obesity and dyslipidemia, which expectedly increases risk of incident CVD. The relationship between HTGW and other metabolic abnormalities in Nigerian hypertensives has not been extensively documented. Our study was aimed at describing the prevalence, patterns, and predictors of metabolic abnormalities in Nigerian hypertensives with the HTGW phenotype.

## Methods

### Study population

This was a cross sectional retrospective study involving the review of the hospital records of adult hypertensives who presented at a Cardiac Centre in Lagos, Nigeria over a 12-year period

(2009 to 2021). A total of 4,412 patients presented at this hospital within the study period and 1,430(32%) were hypertensives, approximately the reported prevalence of hypertension in Nigeria.

## Data collection/measurements

All hypertensives presenting for the first time at the Cardiac Centre are usually extensively evaluated by cardiologists and trained nurses. This evaluation includes their socio-demographic characteristics, lifestyle risk factors, medical history and thorough physical examination. The nurses measure and record their anthropometric indices as well as their blood pressure using standard protocols as described below. These nurses are trained on how to measure these parameters as part of their pre-employment training. After clinical evaluation these patients are requested to undergo laboratory evaluation.

## Anthropometry

Weight and height were measured to the nearest 0.5 kg and 0.1 cm, respectively, with the patients standing and wearing light clothing, no head gear or footwear. Body Mass Index (BMI) was derived as weight in kilograms divided by height$^2$. Waist circumference (WC) was measured with an inelastic tape (to the nearest 0.1 cm), at the level midway between the lower rib margin and the iliac crest.

## Blood pressure measurement

The blood pressure (BP) of the patients was measured by the nurses after five minutes of rest, with the patients seated comfortably, feet on the floor, arm at the level of the heart and free of any constricting clothing. Appropriate–sized cuffs connected to an Omron HEM7233 (Osaka, Japan) digital sphygmomanometer were used in measuring the BP, which was taken initially on both arms, and the arm with the higher value was used in subsequent measurements. Three BP readings were usually taken at two-to three-minutes intervals. The average of three readings was taken for analysis.

## Biochemical parameters

Venous blood sample of all hypertensives were collected after an overnight fasting and stored in the appropriate vacutainer specimen bottles. The samples were transported to the laboratory in thermo-coolers for processing and analysis. Serum concentrations of total cholesterol (TC), triglyceride (TG), low-density lipoprotein cholesterol (LDL-C), high-density lipoprotein cholesterol (HDL-C), fasting plasma glucose (FPG), serum uric acid, and serum creatinine were measured. From the lipid profile Atherogenic Index of Plasma (AIP) was determined as the log transformation of TG/HDL-C. Non-HDL-C was derived as TC minus HDL.

Relevant patient information required for analysis were retrieved from their case notes by trained research assistants. These included:

Age

Sex

Duration of diagnosis of hypertension

History of smoking

History of Alcohol use

Weight

Height

BMI

Waist circumference

Blood Pressure at first visit

The biochemical parameters

These were entered into a spread sheet by the research assistants.

## Ethical approval

Ethics approval for the research was obtained from Health Research Ethics Committee of the Lagos University Teaching Hospital.

## Definition of terms

- *Hypertension: BP ≥ 140/90mmHg or self-volunteered history of hypertension or on treatment for hypertension*

- *BMI was categorized as: Normal < 25kg/m$^2$; Overweight, 25–29.9kg/m$^2$ and Obesity ≥30kg/m$^2$*

- *Enlarged Waist circumference: WC > 94cm in males; > 88cm in females according to the International Federation of Diabetes (IDF)*

- *Diabetes: FPG ≥ 7.0mmol/l, self-volunteered history of DM or on treatment for DM or Elevated HBA1c >6.5%; Impaired Fasting Glucose (IFG) 5.5mmol/l to 6.9mmol/l.*

- *Dyslipidaemia: Serum $T_C$ > 5.2mmol/l; TG > 1.7mmol/l; $LDL_C$ >3.6mmol/l; $HDL_C$ <1.0mmol/l in males; <1.3mmol/l in females; Non-$HDL_C$ >3.37mmol/l; AIP > 0.1*

- *Serum Uric Acid: > 416.4μmol/l in males; > 356.9μmol/l in females*

- *Elevated serum creatinine: Serum creatinine >97.3μmol/l*

*Triglyceridemic and waist phenotypes:*

- *Normal Waist Normal TG (NWNT): WC < 94cm in males; <80cm in females and TG <1.7mmol/l*

- *Enlarged Waist Normal TG(EWNT): WC >94cm in males; >80cm in females and TG <1.7mmol/l*

- *Normal waist Elevated TG (NWET): WC < 94cm in males; <80cm in females and TG >1.7mmol/l*

- *Hypertriglyceridemic Waist (HTGW):): WC >94cm in males; >80cm in females and TG >1.7mmol/l*

## Statistical analysis

Continuous variables were expressed as mean ± standard deviation (SD) or median and interquartile range when skewed, while categorical variables expressed as percentages. Analysis of variance (ANOVA) and Kruskal Wallis were used to compare differences in the 4 phenotypes for the continuous variables while chi square test was used to compare differences between categorical variables. Logistic regression analyses with odds ratios (ORs) at 95% confidence

intervals (CIs) were performed to estimate the association between HTGW phenotype and metabolic abnormalities. All statistical analyses were performed using SPSS version 26.0 software (SPSS Inc, Chicago, IL), and a p < 0.05 was considered as statistically significant. Charts was used for data presentation where appropriate.

## Results

### General characteristics

The records of 1,430 hypertensives were studied but only 582(40.7%) had the complete dataset of interest. Table 1 shows the general characteristics of the population.

There were more males, 309(53.1%) but females were relatively older, p = 0.015. Twenty-five (4.3%) of the hypertensives were smokers while 154(26.5%) used alcohol and were predominantly males, p<0.001 respectively. The mean duration of hypertension was 9.0 (5–16) years. Females tended to have significantly higher mean BMI, were more obese, and had higher mean Tc and LDL-c, while males had significantly higher mean values of TG, VLDL, creatinine, uric acid, and AIP. Blood pressure phenotypes were comparable in both genders.

**Table 1. General characteristics of the study population.**

| Variable | Total | Male | Female | p-value |
|---|---|---|---|---|
| | (n = 582) | (n = 309) | (n = 273) | |
| Age (year) | 56.2 ±13.6 | 54.9 ±13.6 | 57.6 ±13.4 | 0.015* |
| Smoking status (%) | 25 (4.3) | 23 (7.4) | 2 (0.7) | <0.001* |
| Drinking status (%) | 154 (26.5) | 124 (40.1) | 30 (11.0) | <0.001* |
| **Clinical data** | | | | |
| WC (cm) | 105.4 ± 15.6 | 104.3 ± 16.4 | 106.6 ± 14.6 | 0.078 |
| BMI (kg/m$^2$) | 31.7 ± 6.6 | 30.3 ± 5.8 | 33.2 ± 7.1 | <0.001* |
| Overweight (%) | 173 (29.7) | 104 (33.7) | 69 (25.3) | 0.027* |
| Obesity (%) | 338 (58.1) | 160 (51.8) | 178 (65.2) | 0.001* |
| SBP (mmHg) | 154.3 ± 24.3 | 153.0 ± 24.3 | 155.8 ± 24.2 | 0.157 |
| DBP (mmHg) | 88.9 ± 14.9 | 89.7 ± 14.6 | 87.9 ± 15.2 | 0.156 |
| Duration of HTN (yrs) | 9.0(5–16) | 9.0(5–15) | 9.0(5–17) | 0.847 |
| **Laboratory data** | | | | |
| Tc (mmol/l) | 5.17 ± 1.2 | 5.05 ± 1.2 | 5.30 ± 1.1 | 0.011* |
| TG (mmol/l)) | 1.48 ± 0.5 | 1.59 ± 0.5 | 1.34 ± 0.5 | <0.001* |
| LDL (mmol/l) | 3.20 ± 1.1 | 3.10 ± 1.2 | 3.33 ± 0.9 | 0.012* |
| VLDL (mmol/l) | 0.64 ± 0.2 | 0.69 ± 0.2 | 0.59 ± 0.2 | <0.001* |
| HDL (mmol/l) | 1.34 ± 0.3 | 1.29 ± 0.3 | 1.39 ± 0.3 | <0.001* |
| Non-HDL ((mmol) | 3.83 ± 1.1 | 3.77 ± 1.2 | 3.90 ± 1.0 | 0.127 |
| Uric acid (μmol/l) | 429.27 ± 132.5 | 458.48±140.9 | 396.20±113.9 | <0.001* |
| HbA1C (%) | 6.28 ± 1.4 | 6.30 ± 1.5 | 6.30 ± 1.3 | 0.813 |
| FBS (mmol/l) | 6.05 ± 2.1 | 6.01 ± 2.19 | 6.09 ± 2.1 | 0.666 |
| Creatinine (mmol/l) | 97.26 (81–115) | 103.45 (88–124) | 88.42 (77–108) | <0.001* |
| Atherogenic index | 0.9(0.5–1.2) | 1.0(0.6–1.3) | 0.7(0.4–1.0) | <0.001* |

* Statistically significant.

WC, Waist circumference; BMI, Body Mass Index; SBP, Systolic Blood Pressure; DBP, Diastolic Blood Pressure; HTN, Hypertension; Tc, Total cholesterol; TG, Triglyceride; LDL-c, Low density lipoprotein cholesterol; VLDL-c, Very low density lipoprotein cholesterol; HDL-c, High density lipoprotein cholesterol; FBG, Fasting Blood Glucose; HbA1c, Glycosylated hemoglobin; AIP, Atherogenic Index of Plasma.

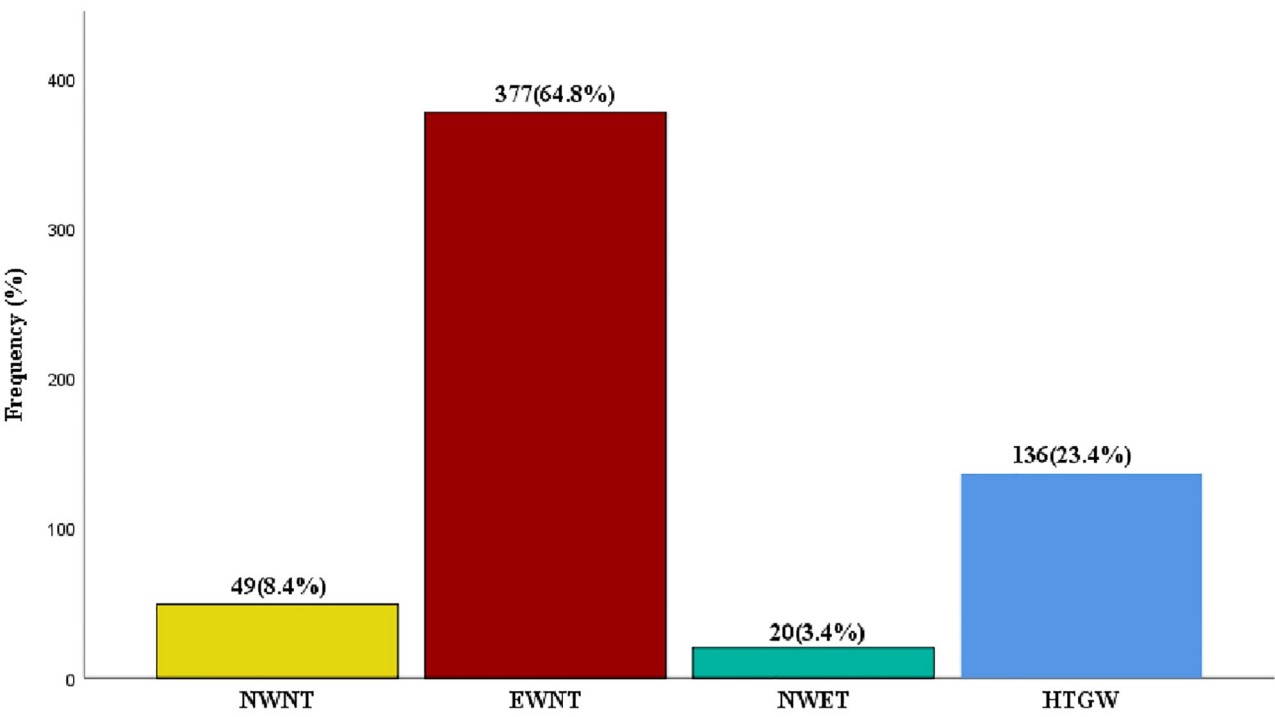

**Fig 1. Distribution of the triglyceridemic-waist phenotype in the study population.**

Majority 484 (83.2%) of hypertensives were on 2 or more medications for BP control. The top three prescribed antihypertensives were Renin Angiotensin System blockers, diuretics, and calcium channel blockers.

## Prevalence and patterns of triglyceridemic-waist phenotype

Fig 1 shows the distribution of the triglyceridemic-waist phenotype. The prevalence of HTGW phenotype was 23.4%.

Table 2 shows the characteristics of the phenotypes. Subjects with the HTGW phenotype were relatively younger, predominantly males (61.0%) and less of smokers compared to NWNT phenotype. The EWNT and HTGW phenotypes had significantly higher mean WC and BMI compared to NWNT and NWET phenotypes, p<0.001 respectively. The prevalence of obesity was comparable in the EWNT and HTGW phenotypes (65.5% vs 63.2%) but significantly higher than in the NWNT and NWET phenotypes, p< 0.001. Mean SBP and DBP values were comparable in all 4 phenotypes, p = 0.33 and 0.92 respectively. The mean value of all biochemical parameters except LDL-c were significantly higher in EWNT, HTGW and NWET phenotypes compared to NWNT phenotypes.

## Prevalence of metabolic perturbations across the triglyceridemic-waist phenotypes

Table 3 shows the metabolic abnormalities across the phenotypes. The HTGW phenotype had higher prevalence of elevated Tc (61.8%), TG (100.0%), VLDL (91.2%), p<0.001; serum uric acid (67.6%) and AIP (100.0%), p = 0.002 and p<0.001 respectively compared to NWNT. Males with HTGW phenotype had significantly higher prevalence of elevated Tc (60.2%), TG (100.0%), VLDL (96.4%) and serum uric acid (63.9%), p = 0.003, and p<0.001 respectively

**Table 2. General characteristics of the hypertensive triglyceridemic-waist phenotypes.**

| Variable | NWNT | EWNT | NWET | HTGW | p-value |
|---|---|---|---|---|---|
| | (n = 49) | (n = 377) | (n = 20) | (n = 136) | |
| Age(yr)± SD | 56.8 ±17.0 | 57.2 ±13.2 | 49.1 ±13.1 | 54.1±12.9 | 0.012* |
| Sex (%) | | | | | <0.001** |
| Male | 43 (87.8) | 165 (43.8) | 18 (90.0) | 83 (61.0) | |
| Female | 6 (12.2) | 212 (56.2) | 2 (10.0) | 53 (39.0) | |
| Smoking status (%) | 3 (6.1) | 13 (3.4) | 3 (15.0) | 6 (4.4) | 0.004** |
| Drinking status (%) | 11 (22.4) | 88 (23.3) | 11 (55.0) | 44 (32.4) | 0.084** |
| **Clinical data** | | | | | |
| WC (cm) | 83.9 ± 11.8 | 108.6 ± 13.6 | 79.2 ± 17.6 | 108.4 ± 11.4 | <0.001* |
| BMI (kg/m$^2$) | 24.1 ± 3.3 | 32.7 ± 6.5 | 25.9 ± 2.9 | 32.3 ± 6.1 | <0.001* |
| Overweight (%) | 14 (28.6) | 107 (28.4 | 9 (45.0) | 43 (31.6) | 0.420 |
| Obesity (%) | 2 (4.1) | 248 (65.8) | 2 (10.0) | 86 (63.2) | <0.001** |
| SBP (mmHg) | 158.3 ± 20.3 | 154.8 ± 23.9 | 149.2 ± 21.9 | 152.2 ± 23.4 | 0.332 |
| DBP (mmHg) | 88.6 ± 16.0 | 88.9 ± 15.0 | 91.1 ± 12.3 | 88.6 ± 14.8 | 0.922 |
| Duration of HTN (year) | 8.0 (5.0–15.0) | 9.0(6.0–17.0) | 8.0(4.0–15.0) | 7.0(5.0–14.0) | 0.053 |
| **Laboratory data** | | | | | |
| Tc (mmol/l) | 4.89 ± 1.1 | 5.08 ± 1.1 | 5.39 ± 1.5 | 5.46 ± 1.1 | 0.003* |
| TG (mmol/l) | 1.12 ± 0.3 | 1.15 ± 0.3 | 2.46 ± 0.7 | 2.36 ± 0.8 | <0.001* |
| LDL (mmol/l) | 3.15 ± 0.9 | 3.15 ± 0.9 | 3.09 ± 0.7 | 3.12 ± 1.1 | 0.592 |
| VLDL (mmol/l) | 0.50 ± 0.1 | 0.50 ± 0.1 | 1.05 ± 0.3 | 1.01 ± 0.3 | <0.001* |
| HDL (mmol/l) | 1.26 ± 0.3 | 1.34 ± 0.3 | 1.25 ± 0.4 | 1.36 ± 0.3 | 0.239 |
| Non-HDL (mmol/l) | 3.63 ± 1.0 | 3.74 ± 1.0 | 4.14 ± 1.7 | 4.11 ± 1.1 | 0.003* |
| Uric acid (μmol/L) | 435.75 ± 114 | 417.10 ± 121 | 514.77 ± 147 | 448.10 ± 109 | 0.002* |
| HbA1C (%) | 6.16 ± 1.3 | 6.20 ±1.4 | 6.74 ± 1.7 | 6.49 ±1.5 | 0.971 |
| FBS (mmol/l) | 6.13 ± 2.0 | 6.00 ± 2.2 | 5.57 ± 2.3 | 6.22 ± 2.1 | 0.599 |
| Creatinine (mmol/l) | 97.26(80–108) | 96.38(80–114) | 110.53(93–213) | 99.03(86–122) | 0.006# |
| Atherogenic index | 0.7(0.4–1.1) | 0.7(0.4–0.9) | 1.5(1.0–1.8) | 1.3(1.1–1.6) | <0.001# |

Asterisk indicate significant;

*ANOVA;

**Chi square test;

# Kruskal wallis test.

WC, Waist circumference; BMI, Body Mass Index; SBP, Systolic Blood Pressure; DBP, Diastolic Blood Pressure; HTN, Hypertension; Tc, Total cholesterol; TG, Triglyceride; LDL-c, Low density lipoprotein cholesterol; VLDL-c, Very low density lipoprotein cholesterol; HDL-c, High density lipoprotein cholesterol; FBG, Fasting Blood Glucose; HbA1c, Glycosylated hemoglobin; AIP, Atherogenic Index of Plasma.

while in the females this phenotype was associated with higher prevalence of diabetes mellitus (39.6%; newly diagnosed), elevated TG (100.0%) and VLDL (83%) levels; p = 0.01 and p<0.001 respectively (Table 3).

## Patterns of metabolic perturbations across the triglyceridemic-waist phenotypes

Fig 2 shows the frequency of the 5 common CVD risk factors found in the study population. Obesity assessed with BMI and WC, dyslipidemia and hyperuricemia were the top three CVD risk factors.

Fig 3 shows the distribution of the number of CVD risk factors in the study population. More than 90% of the study population had more than two risk factors.

**Table 3. Prevalence of metabolic abnormalities in the hypertensive triglyceridemic-waist phenotypes.**

| Variable | Total | NWNT | EWNT | NWET | HTGW | p-value |
|---|---|---|---|---|---|---|
| **Total** | **(n = 582)** | **(n = 49)** | **(n = 377)** | **(n = 20)** | **(n = 136)** | |
| High Tc | 268(46.0) | 18 (36.7) | 158 (41.9) | 8 (40.0) | 84 (61.8) | <0.001* |
| High TG | 156 (26.8) | 0 (0) | 0 (0) | 20 (100) | 136 (100) | <0.001* |
| High LDL | 185 (31.8) | 14 (28.6) | 123 (32.6) | 3 (15.0) | 45 (33.1) | 0.382 |
| High VLDL | 144 (24.7) | 0 (0) | 0 (0) | 20 (100) | 124 (91.2) | <0.001* |
| Low HDL | 97 (16.7) | 13 (26.5) | 61 (16.2) | 6 (30.0) | 17 (12.5) | 0.051 |
| High Non-HDL | 383 (65.8) | 29 (59.2) | 240 (63.7) | 13 (65.0) | 101 (74.3) | 0.109 |
| High Uric acid | 346 (59.5) | 22 (44.9) | 215 (57.0) | 17 (85.0) | 92 (67.6) | 0.002* |
| High Creatinine | 268 (46.0) | 22 (44.9) | 161 (42.7) | 12 (60.)) | 73 (53.7) | 0.091 |
| IGT | 149 (25.6) | 18 (36.7) | 99 (26.3) | 3 (15.0) | 29 (21.3) | 0.124 |
| DM | 123 (21.1) | 9 (18.4) | 72 (19.1) | 3 (15.0) | 39 (28.7) | 0.100 |
| High HbA1C | 198 (33.3) | 17 (34.7) | 119 (31.6) | 8 (40.0) | 50 (36.8) | 0.639 |
| High AI | 554 (95.2) | 45 (91.8) | 353 (93.6) | 20 (100) | 136 (100) | 0.011* |
| **Men** | **(n = 309)** | **(n = 43)** | **(n = 165)** | **(n = 18)** | **(n = 83)** | |
| High Tc | 133 (43.0) | 16 (37.2) | 60 (36.4) | 7 (38.9) | 50 (60.2) | 0.003* |
| High TG | 101 (32.9) | 0 (0) | 0 (0) | 18 (100) | 83 (100) | <0.001* |
| High LDL | 86 (77.8) | 12 (27.9) | 48 (29.1) | 2 (11.1) | 24 (28.9) | 0.443 |
| High VLDL | 98 (31.7) | 0 (0) | 0 (0) | 18 (100) | 80 (96.4) | <0.001* |
| Low HDL | 65 (21.0) | 10 (23.3) | 38 (23.0) | 5 (27.8) | 12 (14.5) | 0.365 |
| High Non-HDL | 191 (61.8) | 25 (58.1) | 95 (57.6) | 18 (100) | 60 (72.3) | 0.147 |
| High Uric acid | 179 (57.9) | 19 (44.2) | 91 (55.2) | 16 (88.9) | 53 (63.9) | 0.007* |
| High Creatinine | 168 (54.4) | 18 (41.9) | 97 (58.8) | 11 (61.1) | 42 (50.6) | 0.186 |
| IGT | 81 (26.2) | 15 (34.9) | 45 (37.3) | 2 (11.1) | 19 (22.9) | 0.225 |
| DM | 60 (19.4) | 7 (16.3) | 32 (19.4) | 3 (16.7) | 18 (21.7) | 0.889 |
| High HbA1C | 102 (33.0) | 14 (32.6) | 54 (32.7) | 7 (38.9) | 27 (32.5) | 0.960 |
| High AI | 295 (95.6) | 40 (93.0) | 154 (93.3) | 18 (100) | 83 (100) | 0.068 |
| **Women** | **(n = 273)** | **(n = 6)** | **(n = 212)** | **(n = 2)** | **(n = 53)** | |
| High Tc | 135 (49.5) | 2 (33.3) | 98 (46.2) | 1 (50.0) | 34 (64.2) | 0.107 |
| High TG | 55 (20.2) | 0 (0) | 0 (0) | 2 (100) | 53 (100) | <0.001* |
| High LDL | 99 (36.3) | 2 (33.3) | 75 (35.4) | 1 (50.0) | 32 (60.4) | 0.915 |
| High VLDL | 46 (16.8) | 0 (0) | 0 (0) | 2 (100) | 44 (83.0) | <0.001* |
| Low HDL | 32 (11.7) | 3 (50.0) | 23 (10.8) | 1 (50.0) | 5 (9.4) | 0.008* |
| High Non-HDL | 192 (70.3) | 4 (66.7) | 145 (68.4) | 2 (100) | 41 (77.4) | 0.472 |
| High Uric acid | 167 (61.2) | 3 (50.0) | 124 (58.5) | 1 (50.0) | 39 (73.6) | 0.212 |
| High Creatinine | 100 (36.6) | 4 (66.7) | 64 (30.2) | 1 (50.0) | 31 (58.5) | 0.001* |
| IGT | 68 (24.9) | 3 (50.0) | 54 (25.5) | 1 (50.0) | 10 (18.9) | 0.288 |
| DM | 63 (23.1) | 2 (33.3) | 40 (18.9) | 0 (0) | 21 (39.6) | 0.010* |
| High HbA1C | 92 (33.7) | 3 (50.0) | 65 (30.7) | 1 (50.0) | 23 (43.4) | 0.255 |
| High AI | 259 (94.9) | 5 (83.3) | 199 (93.9) | 2 (100) | 53 (100) | 0.168 |

Chi square test used.

*Statistically significant.

WC, Waist circumference; BMI, Body Mass Index; SBP, Systolic Blood Pressure; DBP, Diastolic Blood Pressure; HTN, Hypertension; Tc, Total cholesterol; TG, Triglyceride; LDL-c, Low density lipoprotein cholesterol; VLDL-c, Very low density lipoprotein cholesterol; HDL-c, High density lipoprotein cholesterol; FBG, Fasting Blood Glucose; HbA1c, Glycosylated hemoglobin; AIP, Atherogenic Index of Plasma.

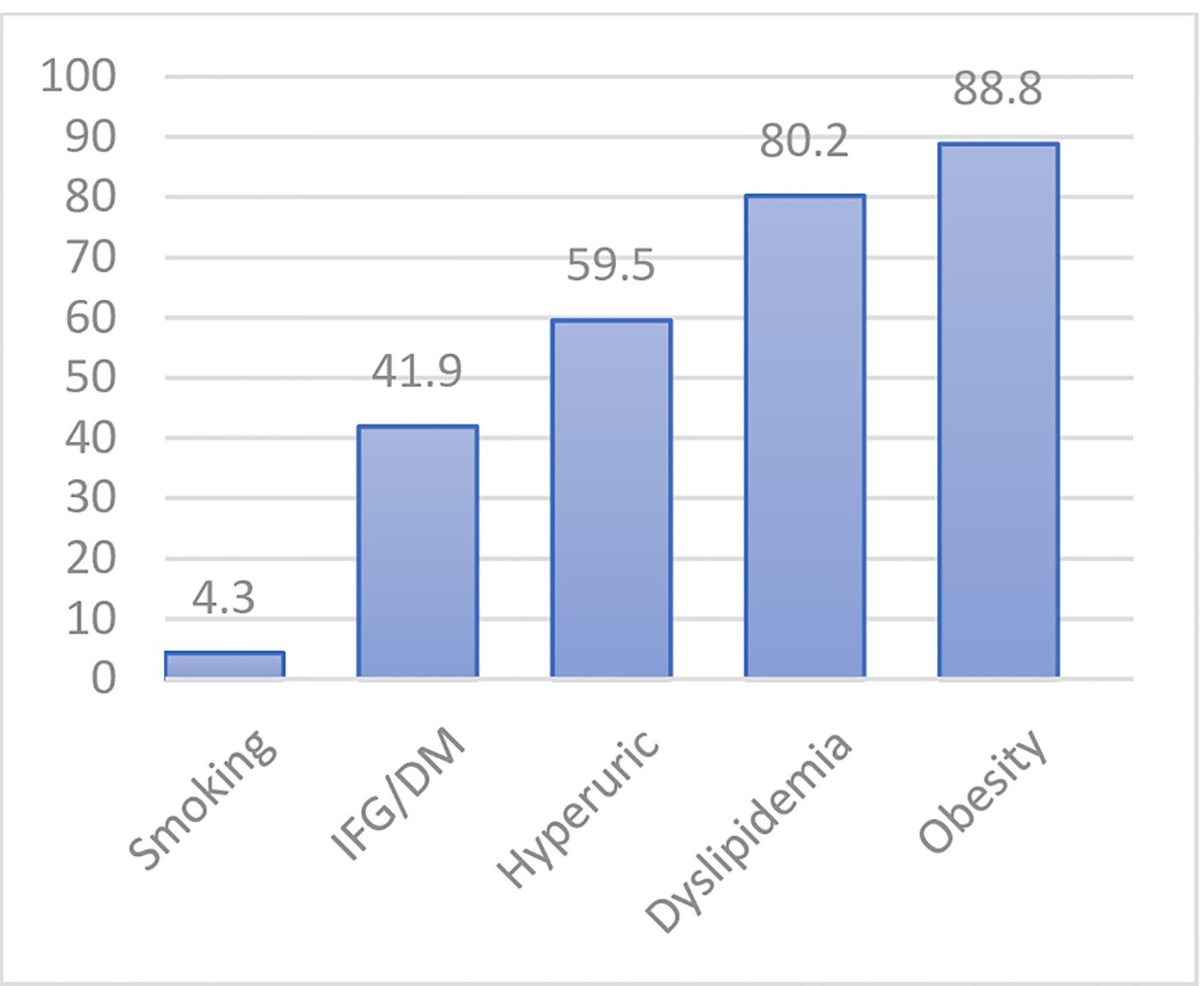

**Fig 2. Prevalence of cardiovascular disease risk factors in the study population.**

Fig 4 shows the frequency of the number of CVD risk factors present in each of the 4 trigly-ceridemic-waist phenotypes. The HTGW phenotype had the highest frequency of co-occurring CVD risk factors of 84.6%.

Fig 5 shows the most frequent CVD risk factor combinations in the study population. The most frequent combinations were obesity, hyperuricemia (25.2%) and obesity, abnormal glucose profile, hyperuricemia, and dyslipidemia (18.5%).

## Predictors of triglyceridemic phenotypes and metabolic abnormalities

Table 4 shows the adjusted odds ratios of the 4 phenotypes and their association with metabolic abnormalities. Compared with the NWNT phenotype the HTGW phenotype had significantly strong associations with elevated Tc, TG, VLDL, uric acid, and AIP in the total population. Subjects (males and females) with the HTGW phenotype were 2.7 times (95% CI, 1.42–5.49; p = 0.003) more likely to have elevated Tc; 3.24 times (95%CI, 1.05–10.39; p<0.001) more likely to have elevated TG; 4.54 times (95% CI, 3.59–7.21; p<0.001) more likely to have elevated VLDL; 2.57 times (95% CI, 1.32–5.00; p = 0.006) for hyperuricemia and 8.8 times

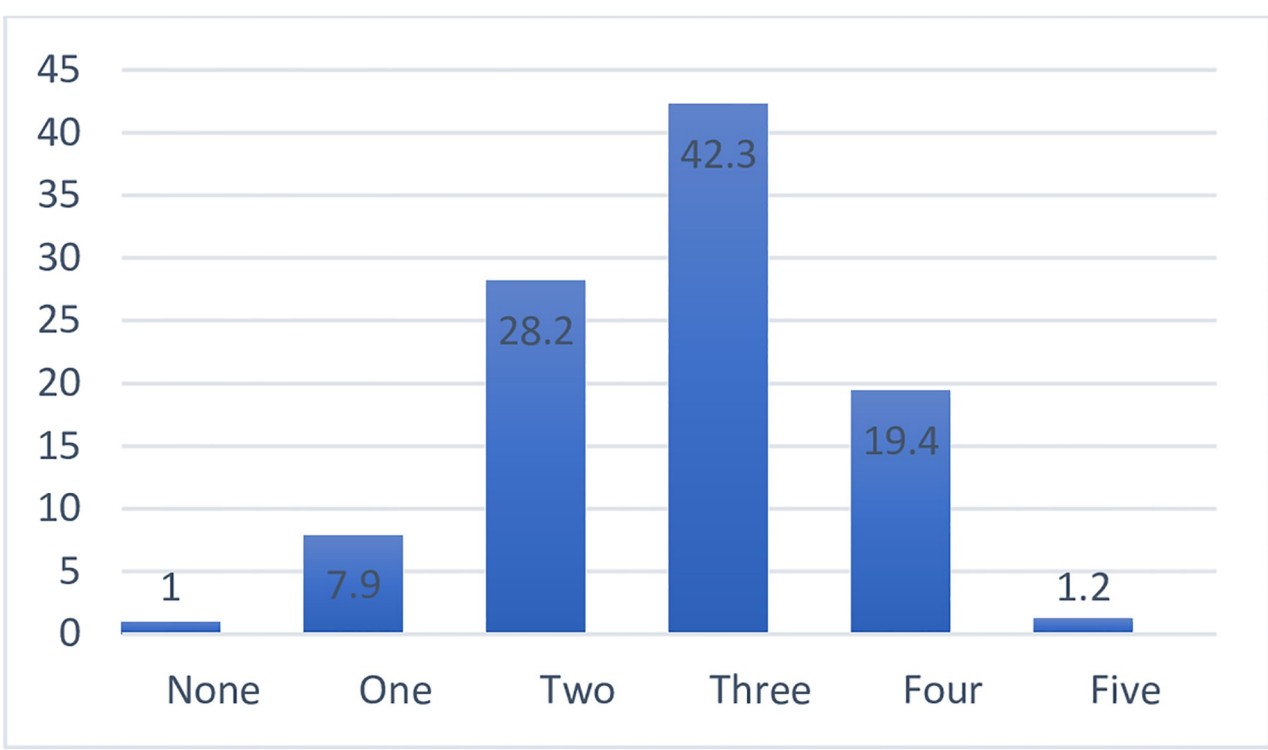

**Fig 3. Distribution of the number of cardiovascular disease risk factors in the study population.**

(95% CI, 0.9–86.6; p = 0.03) of having elevated AIP (Table 4). With respect to elevated VLDL its association was stronger in females than in males; 10.15 (95% CI, 2.70–17.93; p<0.001) vs 4.54 (95% CI, 3.59–7.21; p<0.001 respectively (Table 4).

## Discussion

Although there are many published studies on metabolic syndrome in Nigerians ours is the first (according to the knowledge of the authors) to have explored the relationship between simultaneous presence hypertriglyceridemia and elevated waist circumference (HTGW) and metabolic abnormalities in hypertensives. Our study showed that the prevalence of the HTGW phenotype in our hypertensive cohort was 23.4%; 26.8% and 19.4% in males and females respectively. In the Chinese hypertensive population Chen et al. [15] reported a prevalence of 15.8% and 13.5% and 18.1% in male and female hypertensives respectively. In ELSA-Brazil study the prevalence of HTGW in hypertensives was reported to range between 13.3% and 24.7% with higher prevalence in females [20]. Our prevalence is similar to the 21.4% and 21.5% reported by two other Brazilian studies [14, 21]. The varying prevalence might be related to the different cut-off points used for elevated WC in these studies and the type of population studied. The HTGW phenotype can be likened as a surrogate of metabolic syndrome [22–24]. They both share common criteria especially hypertriglyceridemia and elevated WC and the same cut-off points. In Nigeria the prevalence of metabolic syndrome is 28% using the IDF criteria [25].

Our study reported a higher prevalence of HTGW in males. The impact of gender on the HTGW phenotype in published literature is varied. While some studies reported no gender differences in prevalence [26, 27], other studies reported either male preponderance [22, 28]

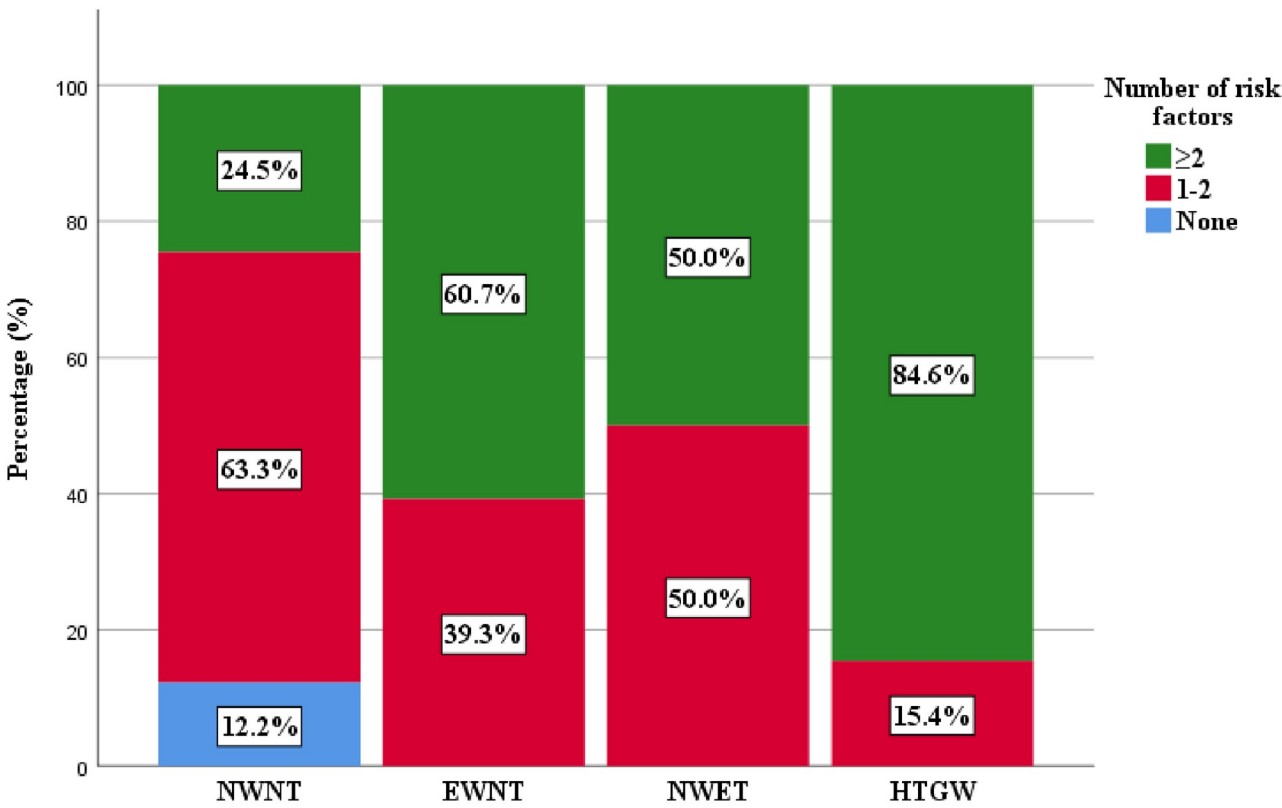

**Fig 4. The frequency of the CVD risk factors in the triglyceridemic-waist phenotypes.**

or female preponderance [14, 15]. These differences might be methodological (gender composition of the study populations and in populations with varying cardiovascular risk factors). In our study we had more males, and this might explain the higher prevalence of HTGW in males.

Our study also demonstrated greater aggregation of metabolic perturbations in the HTGW phenotype, which has been demonstrated by other studies [14–16]. The direct implication of this perturbations is a heightened risk of CVD, prospectively [29–31]. The EPIC-Norfolk study, the largest prospective study to date to explore the relationship between HTGW phenotype and CVD risk, reported of increased risk of CVD and poor outcomes in men and women with the HTGW phenotype compared with those with NWNT phenotype [11]. Other prospective studies in other parts of Europe arrived at this same conclusion [32, 33]. In the Framingham Offspring Study, Wilson et al. [34] reported that hypertriglyceridemic waist contributed to elevated CVD risk and even more to associated with incident Type 2 diabetes mellitus.

The major metabolic abnormalities and the predictors of the found HTGW phenotype in our study were elevated Tc, VLDL, TG, uric acid, and AIP. In males the predictors of the HTGW phenotype were elevated Tc, VLDL-c, uric acid, and AIP. In females only elevated VLDL-c and uric acid were associated with the HTGW phenotype. However, VLDL was a stronger predictor of the HTGW phenotype in females than in males; 10.15 (95% CI, 2.70–17.93; p<0.001) vs 4.54 (95% CI, 3.59–7.48; p<0.001 respectively.

Although the prevalence of elevated high LDL-c was high in both males and females with HTGW phenotype (28.9% vs 60.4% respectively), it was not a predictor of this phenotype. Some studies have reported elevated LDL-c as a predictor of the HTGW phenotype [14–18,

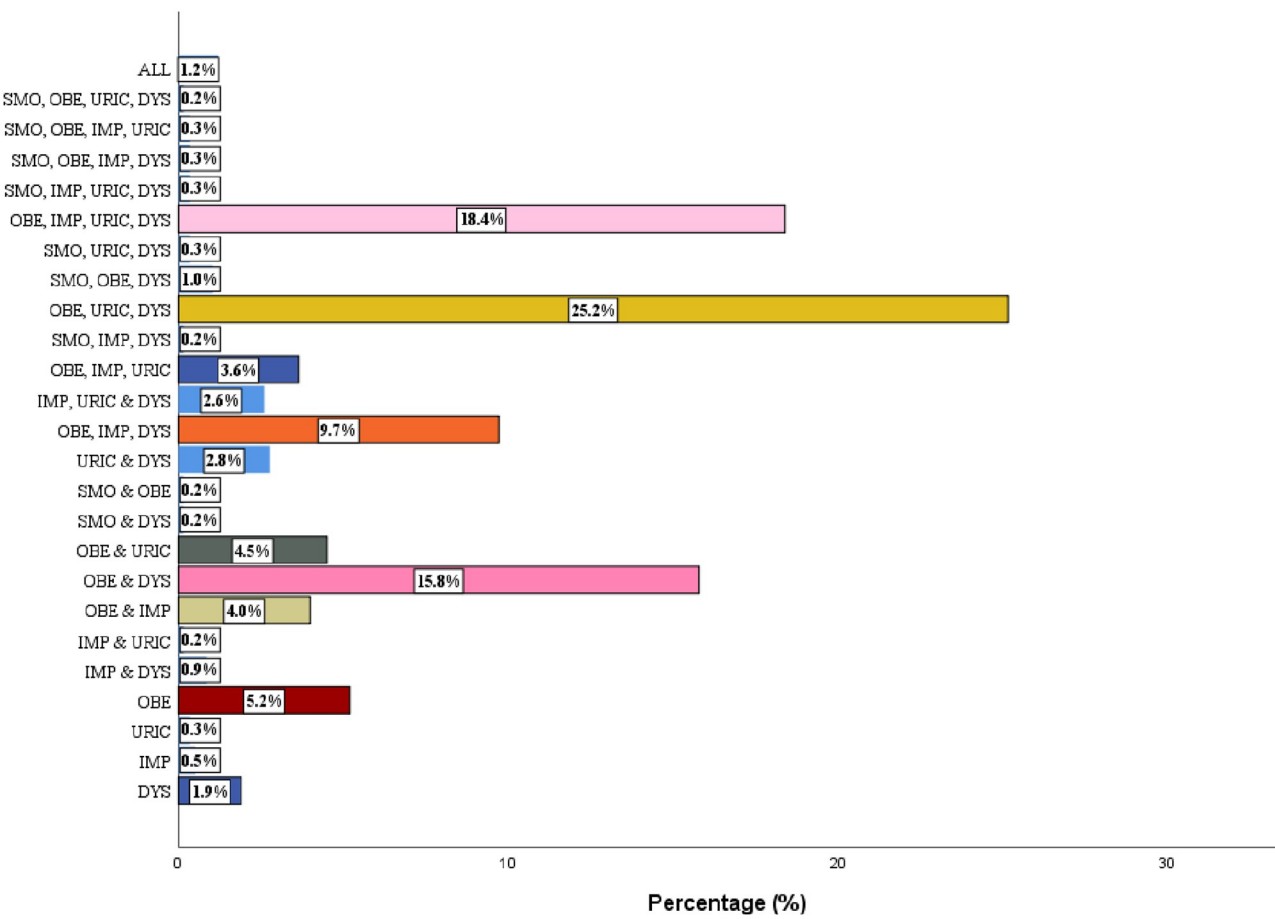

**Fig 5. Some common cardiovascular risk factor combinations in the study population.**

20]. Our HTGW cohort might have been on treatment with statins at the time of enrolment into the study, and this might explain the observed trend in their LDL-c levels. The clinical significance of elevated LDL-c in atherosclerotic CVD is well established. However, serum level of LDL-c does completely and carefully account for atherosclerotic CVD (ASCVD) even as the biological relationship between CVD and dyslipidemia continues to evolve. Very Low-Density Lipoprotein Cholesterol (VLDL-c), a component of non-HDL and triglyceride rich lipid fraction, is an identified risk factor for ASCVD and its importance in the prevention of CVD is widely recognized [34, 35]. Elevated VLDL-c is a major type of dyslipidemia, especially in China, and recent epidemiological studies have demonstrated the superiority of VLDL-c over low-density lipoprotein cholesterol (LDL-c) in terms of the population-attributable risk proportion for ASCVD [36]. Even in the presence of on-target LDL-c levels and in the absence of the traditional risk factors for CVD, elevated VLDL-c levels is associated with 2.19–3.36-fold increased risk of coronary heart disease [36]. Secondly, in the presence of obesity and diabetes (as is the case with our HTGW phenotype) LDL-c is not an accurate marker of CVD risk [37]. Cardiovascular disease events occur in the scenario of on-target or even low levels of LDL-c and/or statin therapy in some individuals, a phenomenon referred to as residual cardiovascular risk [37]. Triglyceride-rich lipoproteins (especially VLDL) and their cholesterol content could partially explain this residual risk [38]. These lipoproteins are novel biomarkers driving residual cardiovascular risk in our contemporary era of high burden of obesity, diabetes, and

**Table 4. Association of Triglyceridemic-waist phenotypes with metabolic abnormalities.**

| Metabolic abnormality | Predictor | Total (n = 582) | | Men (n = 309) | | Women (n = 273) | |
|---|---|---|---|---|---|---|---|
| | | OR (95% CI) | p-value | OR (95% CI) | p-value | OR (95% CI) | p-value |
| High Tc | NWNT | Reference | | Reference | | Reference | |
| | EWNT | 1.24 (9.67–2.30) | 0.489 | 0.96 (0.48–1.93) | 0.918 | 1.72 (0.31–9.59) | 0.537 |
| | NWET | 1.14 (0.40–3.34) | 0.800 | 1.07 (0.35–3.33) | 0.902 | 2.00 (0.08–51.59) | 0.676 |
| | HTGW | 2.78 (1.42–5.49) | 0.003* | 2.26 (1.20–5.46) | 0.015* | 3.58 (0.60–21.39) | 0.162 |
| High TG | NWNT | Reference | | Reference | | Reference | |
| | EWNT | 0.13 (0.01–2.04) | 0.087 | 0.26 (0.02–4.09) | 0.303 | 0.02 (0.00–0.38) | <0.001* |
| | NWET | 9.12 (5.7–14.52) | <0.001* | 3.14 (2.6–11.41) | <0.001* | 0.39 (0.12–2.69) | 0.346 |
| | HTGW | 32.4 (4.05–10.39) | <0.001* | 3.93 (2.15–5.71) | <0.001* | 2.60 (1.7–7.93) | <0.001* |
| High LDL | NWNT | Reference | | Reference | | Reference | |
| | EWNT | 1.21 (0.63–2.33) | 0.568 | 1.06 (0.50–2.24) | 0.874 | 1.10 (0.20–6.12) | 0.918 |
| | NWET | 0.44 (0.11–1.75) | 0.243 | 0.32 (0.06–1.62) | 0.170 | 2.00 (0.08–51.59) | 0.676 |
| | HTGW | 1.24 (0.61–2.53) | 0.561 | 1.05 (0.46–2.38) | 0.905 | 1.31 (0.22–7.82) | 0.765 |
| High VLDL | NWNT | Reference | | Reference | | Reference | |
| | EWNT | 0.13 (0.01–2.04) | 0.087 | 0.26 (0.02–4.09) | 0.303 | 0.20 (0.13–0.38) | <0.001* |
| | NWET | 912 (57–14581) | <0.001* | 7.14 (4.51–11.93) | <0.001* | 0.38 (0.2- | 0.346 |
| | HTGW | 4.54 (3.59–7.21) | <0.001* | 4.54 (3.59–7.21) | <0.001* | 10.15(2.76–17.93) | 0.001* |
| Low HDL | NWNT | Reference | | Reference | | Reference | |
| | EWNT | 0.54 (0.27–1.07) | 0.076 | 0.99 (0.45–2.19) | 0.975 | 0.12 (0.02–0.64) | 0.013* |
| | NWET | 1.19 (0.38–3.74) | 0.770 | 1.27 (0.36–4.45) | 0.709 | 1.00 (0.04–24.55) | 1.000 |
| | HTGW | 0.40 (0.18–0.89) | 0.025* | 0.56 (0.22–1.42) | 0.221 | 0.10 (0.02–0.66) | 0.016* |
| High Non-HDL | NWNT | Reference | | Reference | | Reference | |
| | EWNT | 1.21 (0.66–2.22) | 0.542 | 0.98 (0.50–1.93) | 0.947 | 1.08 (0.19–6.05) | 0.928 |
| | NWET | 1.28 (0.43–3.78) | 0.654 | 1.13 (0.37–3.48) | 0.830 | 0.28 (0.22–0.47) | 0.999 |
| | HTGW | 1.99 (1.00–4.00) | 0.050* | 1.88 (0.87–4.07) | 0.110 | 1.71 (0.28–10.49) | 0.563 |
| Hyperuricemia | NWNT | Reference | | Reference | | Reference | |
| | EWNT | 1.63 (0.90–2.96) | 0.110 | 1.55 (0.79–3.05) | 0.201 | 1.41 (0.28–7.15) | 0.679 |
| | NWET | 7.00 (1.80–26.84) | 0.005* | 10.11 (2.06–49.48) | 0.004* | 1.00 (0.04–24.55) | 1.000 |
| | HTGW | 2.57 (1.32–5.00) | 0.006* | 2.23 (1.05–4.73) | 0.036* | 2.79 (0.50–15.45) | 0.241 |
| High HbA1C | NWNT | Reference | | Reference | | Reference | |
| | EWNT | 0.87 (0.46–1.63) | 0.639 | 1.01 (0.49–2.06) | 0.983 | 0.44 (0.09–2.25) | 0.325 |
| | NWET | 1.26 (0.43–3.66) | 0.678 | 1.32 (0.42–4.13) | 0.636 | 1.00 (0.04–24.55) | 1.000 |
| | HTGW | 1.09 (0.55–2.17) | 0.796 | 0.99 (0.46–2.19) | 0.997 | 0.77 (0.14–4.16) | 0.758 |
| High atherogenic index | NWNT | Reference | | Reference | | Reference | |
| | EWNT | 0.98 (0.702–4.90) | 0.892 | 0.76 (0.16–3.53) | 0.724 | 3.65 (0.47–28.43) | 0.224 |
| | NWET | 1.24 (0.13–12.06) | 0.856 | 2.94 (1.94–4.09) | <0.001* | 0.41 (0.01–1.93) | 0.346 |
| | HTGW | 8.80 (0.90–85.69) | 0.026* | 4.01 (3.11–5.79) | <0.01* | 10.40 (0.65–16.23) | 0.058 |

* Statistically significant.

WC, Waist circumference; BMI, Body Mass Index; SBP, Systolic Blood Pressure; DBP, Diastolic Blood Pressure; HTN, Hypertension; Tc, Total cholesterol; TG, Triglyceride; LDL-c, Low density lipoprotein cholesterol; VLDL-c, Very low density lipoprotein cholesterol; HDL-c, High density lipoprotein cholesterol; FBG, Fasting Blood Glucose; HbA1c, Glycosylated hemoglobin; AIP, Atherogenic Index of Plasma.

metabolic syndrome [38, 39]. The JUPITER trial demonstrated that VLDL-c, particularly the smallest remnant subclass, was associated with cardiovascular disease risk when LDL was low [40]. VLDL remnants (as well as LDL particles) have been shown to migrate across the endothelium where they are entrapped by macrophages, forming foam cells, promoting low-grade inflammation, and facilitating atheromatous plaque growth [41].

Hyperuricemia is strongly associated with CVD and CVD risk [42, 43]. In addition, this relationship has a J-curve phenomenon [44]. It is positively associated with obesity, hypertension, and dyslipidemia, and hyperuricemic subjects tend to have a clustering of these cardiovascular risk factors which also are components of the metabolic syndrome [45]. Viscerally obese people are known to produce high levels of uric acid. Our HTGW subjects had a significantly higher prevalence of hyperuricemia (67.6%) compared to those with NWNT phenotype (44.9%). This corroborates findings from similar studies [16, 27, 28]. The high prevalence of hyperuricemia in the HTGW phenotype agrees with the reported finding that elevated serum uric acid is associated with the metabolic syndrome, of which the HTGW phenotype is a surrogate of [46–48]. This clustering of hyperuricemia with metabolic syndrome or its components is associated with increased CVD risk. Our study also reported prevalence of 59.5% for hyperuricemia in all subjects irrespective of their phenotypes. In Nigeria the prevalence of hyperuricemia is higher in hypertensives with prevalence rates ranging between 36.7% and 60.2% compared to 17.2% to 20.5% in the general population [49–51]. Hyperuricemia and hypertension have a bi-directional relationship. The former is more prevalent in hypertensives and is also a risk factor for incident hypertension. The association between serum uric acid levels and high blood pressure in humans is well established. For example, cross-sectional and longitudinal studies have shown that there is a 13% increase in the risk of incident hypertension for each 1 mg/dL increase in serum uric acid in a general normotensive population not treated for hyperuricemia [52, 53]. This association is linear and commoner in the younger population and in females [54]. Moreover, hyperuricemia also contributes to the development of hypertension from prehypertension although no causality has been established between hyperuricemia and hypertension [55–57]. Possible mechanisms for the hyperuricemia and hypertension relationship include activation of the intra-renal renin-angiotensin system, urate deposition in the lumen of the nephrons and inflammation [56, 58, 59]. In our study hyperuricemia was a strong predictor of the HTGW phenotype especially in females.

Our study was able to demonstrate high prevalence of the hypertriglyceridemic waist phenotype in a cross section of Nigerian hypertensives and this phenotype is associated with multiple co-occurring metabolic perturbations that are likely to drive the risk of incident CVD. The triglyceridemic-waist is an inexpensive and simple to measure clinical parameter that can add a lot of value to assessing the cardiovascular risk of the hypertensives.

Our study has a few limitations. First it is a cross sectional study and will not be able to prove causality. Secondly the strength of association from this cross-sectional study cannot be as strong as a longitudinal study.

## Supporting information

**S1 Data. HTGW data for statistical analysis.**
(XLSX)

**S2 Data. HTGW raw data.**
(XLSX)

## Author Contributions

**Conceptualization:** Casmir E. Amadi, Amam C. Mbakwem.

**Data curation:** Casmir E. Amadi, Dolapo C. Duro, Clement M. Akinsola.

**Formal analysis:** Dolapo C. Duro, Clement M. Akinsola.

**Supervision:** Amam C. Mbakwem, Ifeoma C. Udenze, Jayne N. Ajuluchukwu, David A. Wale Oke.

**Writing – original draft:** Casmir E. Amadi.

**Writing – review & editing:** Casmir E. Amadi, Ifeoma C. Udenze, Jayne N. Ajuluchukwu, David A. Wale Oke.

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
