## [Decision Letter · Decision Letter 0]

26 Aug 2022

PGPH-D-22-01198

Prevalence, Patterns and Predictors of metabolic abnormalities in Nigerian hypertensives with hypertriglyceridemic waist phenotype: a cross sectional study

Dear Dr. Amadi,

Thank you for submitting your manuscript to PLOS Global Public Health. After careful consideration, we feel that it has merit but does not fully meet PLOS Global Public Health’s publication criteria as it currently stands. Therefore, we invite you to submit a revised version of the manuscript that addresses the points raised during the review process.

We look forward to receiving your revised manuscript.

Kind regards,

Maurizio Trevisan, M.D., MS

Academic Editor

Journal Requirements:

1. In the online submission form, you indicated that "Data is available on request". All PLOS journals now require all data underlying the findings described in their manuscript to be freely available to other researchers, either 1. In a public repository, 2. Within the manuscript itself, or 3. Uploaded as supplementary information.

2. "Please provide separate figure files in .tif or .eps format.

Additional Editor Comments (if provided):

Reviewers' comments:

Reviewer's Responses to Questions

**Comments to the Author**

1. Does this manuscript meet PLOS Global Public Health’s publication criteria? Is the manuscript technically sound, and do the data support the conclusions? The manuscript must describe methodologically and ethically rigorous research with conclusions that are appropriately drawn based on the data presented.

Reviewer #1: Partly

2. Has the statistical analysis been performed appropriately and rigorously?

Reviewer #1: Yes

3. Have the authors made all data underlying the findings in their manuscript fully available (please refer to the Data Availability Statement at the start of the manuscript PDF file)?

Reviewer #1: Yes

4. Is the manuscript presented in an intelligible fashion and written in standard English?

Reviewer #1: Yes

5. Review Comments to the Author

Reviewer #1: The manuscript provides good information about the use of waist and circumference and hypertriglyceridemia as a simple and low-cost measure of visceral obesity and its association with cardio-metabolic abnormalities.

In the presentation of the adjusted odd ratios (OR) in Table 4, I would like to recommend the authors conduct a thorough review of the table as it seems there are some inconsistencies with the OR presented when the Total is broken-down by sex. The sex variable is binary (male and female), and there are huge differences with the OR by sex when compared to the total, for example:

Metabolic abnormality Predictor Total Male Female

High TG NWET 9.12 3.14 0.39

HTGW 32.4 3.93 2.60

High VLDL NWET 912 7.14 0.38

From which table do the data come? With too many variables in the analysis, sometimes the presentation of the results and discussion is challenging to follow the main highlights of the manuscript. I would like to recommend that, when possible, the authors refer to which table the results belong, as sometimes there is no reference to the table. I would also like to recommend that authors consider just selecting the most relevant findings rather than trying to review all of them; otherwise, with the use of the acronyms for each of the variables, the reader may lose the most relevant issue. For example: “Our HTGW cohort also had a high prevalence of diabetes mellitus in both males and females (21.7% and 39.6% respectively) compared to NWNT phenotype, though significant only in the female.”

Finally, it appears that ANOVA results are the center of the presentation of the results and discussion; however, the tables do not refer to the type of analysis that led to the data presented in each of them. OR seems to have valuable information, but they are hardly referred to in the text. When presenting the OR information, I would like to recommend being more specific, for example: “The association between elevated VLDL-c was stronger in females, based on ORs.” Interesting comments, but unless I review Table 4, I will not be able to draw any conclusion about the strength of the association.

6. PLOS authors have the option to publish the peer review history of their article (what does this mean?). If published, this will include your full peer review and any attached files.

**Do you want your identity to be public for this peer review?** For information about this choice, including consent withdrawal, please see our Privacy Policy.

Reviewer #1: No

---

## [Decision Letter · Decision Letter 1]

28 Nov 2022

Prevalence, Patterns and Predictors of metabolic abnormalities in Nigerian hypertensives with hypertriglyceridemic waist phenotype: a cross sectional study

PGPH-D-22-01198R1

Dear Dr Amadi,

We are pleased to inform you that your manuscript 'Prevalence, Patterns and Predictors of metabolic abnormalities in Nigerian hypertensives with hypertriglyceridemic waist phenotype: a cross sectional study' has been provisionally accepted for publication in PLOS Global Public Health.

Best regards,

Julia Robinson

Executive Editor

Reviewer Comments (if any, and for reference):

Reviewer's Responses to Questions

**Comments to the Author**

1. If the authors have adequately addressed your comments raised in a previous round of review and you feel that this manuscript is now acceptable for publication, you may indicate that here to bypass the “Comments to the Author” section, enter your conflict of interest statement in the “Confidential to Editor” section, and submit your "Accept" recommendation.

Reviewer #1: All comments have been addressed

2. Does this manuscript meet PLOS Global Public Health’s publication criteria? Is the manuscript technically sound, and do the data support the conclusions? The manuscript must describe methodologically and ethically rigorous research with conclusions that are appropriately drawn based on the data presented.

Reviewer #1: Yes

3. Has the statistical analysis been performed appropriately and rigorously?

Reviewer #1: Yes

4. Have the authors made all data underlying the findings in their manuscript fully available (please refer to the Data Availability Statement at the start of the manuscript PDF file)?

Reviewer #1: Yes

5. Is the manuscript presented in an intelligible fashion and written in standard English?

Reviewer #1: Yes

6. Review Comments to the Author

Reviewer #1: the manuscript address my first review comments. I recommend its publications

7. PLOS authors have the option to publish the peer review history of their article (what does this mean?). If published, this will include your full peer review and any attached files.

**Do you want your identity to be public for this peer review?** For information about this choice, including consent withdrawal, please see our Privacy Policy.

Reviewer #1: **Yes: **Federico G de Cosio
